# Fully Convolutional Mesh Autoencoder using Efficient Spatially Varying Kernels

**Yi Zhou**[*]
Adobe Research

**Chenglei Wu**
Facebook Reality Labs

**Zimo Li**
University of Southern California

**Chen Cao**
Facebook Reality Labs

**Yuting Ye**
Facebook Reality Labs

**Jason Saragih**
Facebook Reality Labs

**Hao Li**
Pinscreen

**Yaser Sheikh**
Facebook Reality Labs

## Abstract

Learning latent representations of registered meshes is useful for many 3D tasks. Techniques have recently shifted to neural mesh autoencoders. Although they demonstrate higher precision than traditional methods, they remain unable to capture fine-grained deformations. Furthermore, these methods can only be applied to a template-specific surface mesh, and is not applicable to more general meshes, like tetrahedrons and non-manifold meshes. While more general graph convolution methods can be employed, they lack performance in reconstruction precision and require higher memory usage. In this paper, we propose a non-template-specific fully convolutional mesh autoencoder for arbitrary registered mesh data. It is enabled by our novel convolution and (un)pooling operators learned with globally shared weights and locally varying coefficients which can efficiently capture the spatially varying contents presented by irregular mesh connections. Our model outperforms state-of-the-art methods on reconstruction accuracy. In addition, the latent codes of our network are fully localized thanks to the fully convolutional structure, and thus have much higher interpolation capability than many traditional 3D mesh generation models.

## 1 Introduction

Learning latent representations for registered meshes [2], either from performance capture or physical simulation, is a core component for many 3D tasks, ranging from compressing and reconstruction to animation and simulation. While in the past, principal component analysis (PCA) models [1, 21, 30, 35] or manually defined blendshapes [31, 6, 20] have been employed to construct a linear latent space for 3D mesh data, recent works tend towards deep learning models. These models are able to produce more descriptive latent spaces, useful for capturing details like cloth wrinkles or facial expressions.

Convolutional neural networks (CNN) are widely used to capture the spatial features in regular grids, but due to the irregular sampling and connections in the mesh data, spatially-shared convolution kernels cannot be directly applied on meshes as in regular 2D or 3D grid data. A common compromised approach is to first map the 3D mesh data to a predefined UV space, and then train a classic 2D CNN to learn features in the UV space. However, this will inevitably suffer from the parameterization

---

[*]Work partially done during an internship at Facebook Reality Labs, Pittsburgh.
[2]Registered meshes are deformable meshes with a fixed topology.

distortion and the seam/cut in the UV to warp a watertight mesh to a 2D image (See Figure **??** in the appendix), not to mention that this work-around cannot be easily extended to more general mesh data, like tetrahedron meshes.

As a more elegant approach, convolutional neural networks (CNN) designed directly for meshes or graphs were utilized in 3D autoencoders (AE) to achieve state-of-the-art (SOTA) results. Ranjan et al. [26] proposed to use spectral convolution layers and quadric mesh up-and-down sampling methods for constructing a mesh autoencoder called CoMA, and achieved promising results in aligned 3D face data. However, the spectral CNN is not adept at learning data with greater global variations and suffers from oscillation problems. To solve that, Bouritsas et al. [7] replaced the spectral convolution layer by a novel spiral convolution operator and their named Neural3DMM model achieved the state-of-the-art accuracy for both 3D aligned face data and aligned human body motion data. However, both of these methods only work for 2-manifold meshes, and still don't achieve the precision necessary to capture fine-grained deformations or motion in the data.On the other hand, cutting-edge graph convolution operators like GAT [32], MoNet [25] and FeastConv [33], although capable of being applied on general mesh data, exhibit much worse performance for accurately encoding and decoding the vertices' 3D positions.

One major challenge in developing these non-spectral methods is to define an operator that works with different numbers of neighbors, yet maintains the weight sharing property of CNNs. It is also necessary to enable transpose convolution and unpooling layers which allow for compression and reconstruction.

With these in mind, we propose the first template-free fully-convolutional autoencoder for arbitrary registered meshes, like tetrahedrons and non-manifold meshes. The autoencoder has a fully convolutional architecture empowered by our novel mesh convolution operators and (un)pooling operators.

One key feature of our method is the use a spatially-varying convolution kernel which accounts for irregular sampling and connectivity in the dataset. In simpler terms, every vertex will have its own convolution kernel. While a naive implementation of a different kernel at every vertex location can be memory intensive, we instead estimate these local kernels by sampling from a global kernel weight basis. By jointly learning the global kernel weight basis, and a low dimensional sampling function for each individual kernel, we greatly reduce the number of parameters compared to a naive implementation.

In our experiments, we demonstrate that both our proposed AE and the convolution and (un)pooling operators exceed SOTA performance on the D-FAUST [4] dynamic 3D human body dataset which contains large variations in both pose and local details. Our model is also the first mesh autoencoder that demonstrates the ability to highly compress and reconstruct high resolution meshes as well as simulation data in the form of tetrahedrons or non-manifold meshes. In addition, our fully convolutional autoencoder architecture has the advantage of semantically meaningful localized latent codes, which enables better semantic interpolation and artistic manipulation than that with global latent features. From our knowledge, we are the first mesh AE that can achieve localized interpolation.

## 2    Related Work

### 2.1    Deep Learning Efforts on 3D Mesh Autoencoders

A series of efforts have been put to train deep neural auto-encoders to achieve better latent representations and higher generation and reconstruction power than traditional linear methods. Due to the irregularity of local structures in meshes (varying vertex degree, varying sampling density, etc), ordinary 2D or 3D convolution networks cannot be directly applied on mesh data. Thus, early methods attempted to train 2D CNNs on geometry images obtained by parameterizing the 3D meshes to uv space [2] or to spheres as in [27, 28] or torus [23] first and then with proper cutting and unfolding to the 2D image. This type of method is useful in handling high-resolution meshes but suffers from artifacts brought by distortion and discontinuity along the seams. Other types of works [30] simply learning the parameters of an off the rack PCA model.

Later works sought for convolution operators on meshes directly. Litany et al.  [22] proposed a mesh VAE for registered 3D meshes where they stacked several graph convolution layers at the two ends of an auto-encoder. However, the convolution layers didn't function for mesh dimensionality

reduction and accretion, so the latent code was obtained in a rough way of averaging all vertices' feature vectors out. Ranjan et al. [26] proposed a mesh AE called CoMA which has the multi-scale hierarchical structure. The network used spectral convolution layers with Chebychev basis [10] as filters and quadric mesh simplification and its reverse for down and up-sampling. The quadric scaling parameters are determined by a template mesh and shared and fixed for all other meshes.

State of the art work Neural3DMM [7] replaces the spectral convolution layers with operators that convolve along a spiral path around each vertex, outperforming CoMA and the other previous works. The starting point of the spiral path is determined by the geometry of the template mesh. Despite the promising results on surface meshes, both CoMA and Neural3DMM are not applicable to more general mesh types like volumetric and non-manifold meshes. Moreover, the Quadric Mesh sampling is based more on the Euclidean distance of the vertices on the template mesh than the mesh's actual topology, thus, leading to discontinuous receptive fields of the network neurons that break the locality of latent features in the bottleneck layer.

## 2.2 Graph Convolution Networks

In addition to the above mentioned works, there are also many other graph or mesh convolution operators widely utilized in networks for tasks like classification, segmentation and correspondence learning.

Some of them are limited to triangular surface meshes. Masci et al. designed GCNN [24] which uses locally constructed geodesic polar coordinates. Hanocka et al. [14] developed MeshCNN with edge-based convolution operators and scaling layers by leveraging their intrinsic geodesic connections. The defined edge-convolution is direction invariant and so cannot be directly applied for reconstructions. Furthermore, as edges outnumber vertices, this method is not ideal for high resolution data.

Some graph convolution operators are unsuitable for reconstructions because they rely on the vertices' 3D coordinates as input. Fey et al. [12] proposed to use a spline for kernel generation (SplineCNN) which requires inputting pseudo-coordinates computed by the 3D positions of the vertices. Huang et al. [16] designed TextureNet, which defines a 4-rotational symmetric field based on local texture parameterization. In terms of handling varying sampling density of mesh vertices, Hermosilla et al.[15] proposed to use Monta Carlo sampling for non-uniformly sampled point clouds where vertices are weighted by the local density estimated based on their 3D positions. However, this only works in cases where the features for computing densities are not the prediction target, e.g. point cloud segmentation, but not for reconstruction, as our method is designed for.

Spectral approaches [8, 11, 18, 10] work on spectral representations of graphs. While this domain adaptation allows us to leverage the power of standard CNNs directly, it also loses the fidelity of the original signal and the ability to learn directly in the spatial domain. This leads to lower precision in the case of generative models, especially for the task of reconstruction. Other trends, including Graph Attention GAT [32], ANN [5], MoNet [25] and FeastNet [33], take local input vertex features to compute the attention weights with either different types of kernels or learnable functions. Many 3D point cloud and mesh learning methods [34, 22] utilize these graph convolution layers in their models.

While many of those methods model the local variations based on the input features, our method differs from these in that our local convolution is independent from input features and fully learned to be shared across all the training samples. Ours are on the design choice that we want to have local coefficients to only model the graph irregularity, like varying sampling density and orders, while the feature statistics are learned in the shared weight bases.

It is important to note that all graph convolution methods listed above do not support transpose convolution, thus in tasks that requires up-sampling, one needs to apply additional unpooling layers, while our method does enable transpose convolution layers.

## 3 Method

In this section, we first introduce our convolution and transpose convolution operations, named as **vcConv** and **vcTransConv**. We assume the convolution weights lie in the span of a shared weight basis, and are sampled by a set of local coefficients per neighbor. These local coefficients are called

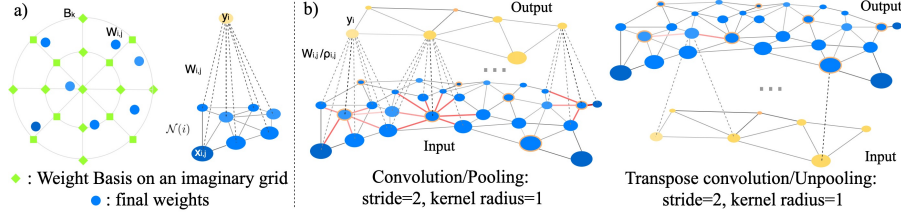

Figure 1: a) Convolution weights. b) Graph sampling.

"variant coefficients"(vc). We call our pooling and unpooling layers **vdPool** and **vdUnpool** since we propose to weight the input vertices with learned densities, which accounts better for the irregularity during rescaling, and those density weights are called "variant densities"(vd). We design residual layers **vcUpRes** and **vcDownRes** in a similar manner respectively for up and down-sampling. Finally, we construct a fully convolutional mesh autoencoder using these proposed graph layers. It has the ability to locally interpolate the mesh and supports user manipulation.

### 3.1  vcConv and vcTransConv

Suppose an output graph $\mathcal{Y}$ is sampled from an input graph $\mathcal{X}$. $\mathcal{Y}$ has $N$ vertices, and each vertex $y_i$ is computed from a local region $\mathcal{N}(i)$ in $\mathcal{X}$. $\mathcal{N}(i)$ has $E_i$ vertices $x_{i,j}, j = 1, .., E_i$.

In a 2D convolution operation, output feature $\mathbf{y}_i \in \mathbb{R}^O$ is computed as:

$$\mathbf{y}_i = \sum_{\mathbf{x}_{i,j} \in \mathcal{N}(i)} \mathbf{W}_j^T \mathbf{x}_{i,j} + \mathbf{b} \tag{1}$$

, where $\mathbf{x}_{i,j} \in \mathbb{R}^I$ are the input feature, $\mathbf{W}_j \in \mathbb{R}^{I \times O}$ is the learned weight matrix defined for each neighboring vertex, and $\mathbf{b}$ is the learned bias. In a regular grid, as the topology of all neighborhoods are identical, $W_j$ can be defined consistently and shared within all the vertices in the grid.

However, on a mesh, vertices are unevenly distributed in the space, and each vertex has different connectivity and direction, so the same weighting schemes cannot be directly applied. One solution for this could be to allow the weights to vary spatially [29], so that each vertex freely defines its own convolution weights. This is called locally connected convolution (LCConv). An LCConv layer has $IO \sum_{i=1}^{N} E_i + O$ training parameters and will require considerable memory. This over-parameterization is also prone to overfitting.

In this paper, we propose a much more efficient spatially varying convolution method. The intuition is that, as illustrated as a 2D case in Figure 1a, we can imagine a discrete convolution kernel defined with weights on a standard grid and we call them Weight Basis. The real vertices in a local region of the mesh scatter in the grid. The Weight Basis can be shared through the whole mesh, while the weights on those real vertices need to be sampled from the Weight Basis by different functions from vertex to vertex. Another perspective for this intuition is that since a mesh is a discretization of a continuous space and a continuous convolution kernel can be shared spatially on the original continuous space, we should be able to resample the unique continuous kernel to generate the weights for each neighborhood of the vertex. To achieve that, the sampling functions need to be defined per vertex locally. Rather than using a handcrafted sampling functions , we learn them through training.

Specifically, we compute the weights per each $x_{i,j}$ as the linear combination of the Weight Basis $B = \{\mathbf{B}_k\}_{k=1}^{M}, \mathbf{B}_k \in \mathbb{R}^{I \times O}$ with locally variant coefficients(vc) $A_{i,j} = \{\alpha_{i,j,k}\}_{k=1}^{M}, \alpha \in \mathbb{R}$:

$$\mathbf{W}_{i,j} = \sum_{k=1}^{M} \alpha_{i,j,k} \mathbf{B}_k \tag{2}$$

, and compute the convolution as

$$\mathbf{y}_i = \sum_{x_{i,j} \in \mathcal{N}(i)} \mathbf{W}_{i,j}^T \mathbf{x}_{i,j} + \mathbf{b} \tag{3}$$

$A_{i,j}$ are different for each vertex $x_{i,j}$ in $\mathcal{N}(i)$ of each $y_i$, but $B$ is shared globally in one layer. They are both learnable parameters and shared across the entire dataset. To remove the scaling

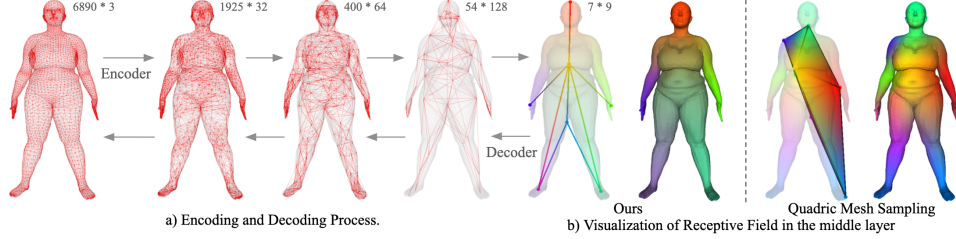

a) Encoding and Decoding Process.

Ours         Quadric Mesh Sampling

b) Visualization of Receptive Field in the middle layer

Figure 2: Network architecture and comparison for receptive fields.

ambiguity between the weight basis and variant coefficients, one can additionally normalize $\mathbf{B}_k$ before multiplying with $\alpha$. However, we found in our ablation study that this can slow down the convergence and lead to higher error.

With this formulation, our parameter count is reduced to $IOM + M \sum_{i=1}^{N} E_i + O$. The size of the Weight Basis determines the network approximation capability. Empirically, we choose $M$ to be roughly the average size of a neighborhood. Section 4.4 shows vcConv uses fewer parameters and less memory than LCConv.

While existing graph convolution operators don't support both down and up scaling functionality, we provide convolution and transpose convolution operators with stride and radius control as in standard 2D convolutions. As illustrated in Figure 1b, the up and down samplings are dual operations but allows for different radius and learning parameters. The sampling is totally based on the mesh's topology so are the same for the whole training set. Manual assignment of certain vertices are allowed for specific purposes. Details of the algorithm can be found in the appendix.

### 3.2 vdPool, vdUnpool, vdUpRes, and vdDownRes

To be consistent with traditional CNNs, we also need to define our Pool and Unpool layers. Naively, we could use max or average operations, which work great for regular grids. However, in an arbitrary graph, the vertices can distribute quite unevenly within the kernel radius, and our experiments in Section 4.4 show that simply using max or average pooling doesn't perform well.

Inspired by Hermosilla et al.[15], we apply Monte Carlo sampling for feature aggregation. In [15], the vertex density is estimated by the 3D coordinates of its neighboring vertices. However, in more general cases, we don't have such information for each layer. While it's hard to design a generally rational density estimation function, we let the network learn the optimal variant density (vd) coefficients across all the training samples. Note that vd is defined per node after pooling or unpooling.

Specifically, the aggregation functions in vdPool and vdUnpool layers are

$$M\mathbf{y}_i = \sum_{j \in \mathcal{N}(i)} \rho'_{i,j}\mathbf{x}_{i,j}, \quad \rho'_{i,j} = \frac{|\rho_{i,j}|}{\sum_{j=1}^{E_i} |\rho_{i,j}|} \tag{4}$$

, where $\rho_{i,j} \in \mathbb{R}$ is the training parameter and $\rho'_{i,j}$ is the density value. Due to the vd coefficient normalization, the vdPool/vdUnpool does not perform any rescaling nor change the mean values of the input feature map.

Similarly, we can define a residual layer as:

$$\mathbf{y}_i = \sum_{x_{i,j} \in \mathcal{N}(i)} \rho'_{i,j}\mathbf{C}\mathbf{x}_{i,j} \tag{5}$$

When the input and output feature dimensions are the same, $\mathbf{C}$ is an identity matrix, otherwise, $\mathbf{C}$ is a learned $O \times I$ matrix shared across all the graph nodes.

With the residual layer, we can design a residual block for up or down-sampling. As illustrated in Figure **??** in the appendix, the input passes through the vcConv or vcTransConv layer and the activation layer Elu [9], and is then added by the output of the vdDownRes or vdUpRes layer. The convolution and residual layer should have the same sampling stride. We denote it as vcConv+vdDownRes or vcTransConv+vdUpRes. For simplicity, we don't denote Elu in the rest of the paper.

### 3.3 Fully Convolutional Autoencoder

Based on the operators above, we propose a fully convolutional mesh AE. Different from [7, 26, 19], ours has no fully connected layers in the middle and is purely built with residual blocks. Figure 2a shows the architecture of an AE on D-FAUST meshes. The network has four down-sampling blocks and four up-sampling blocks with stride $s = 2$ and kernel radius $r = 2$ for all layers. It compresses the original mesh to 7 vertices, 9 channels per vertex, resulting in a 63 dimensional latent code.

**Localized Latent feature Interpolation.** Our graph sampling scheme allows for automatic or manual choice of latent vertices. Here we place the latent vertices on the head, hands, feet, and torso. Their receptive fields will naturally centralize at these vertices and propagate gradually on the surface as visualized in Figure 2b. Consequently, the latent code defines a semantically meaningful latent space. For instance, we can interpolate only the latent vertex on the right arm between a source and a target code, to alter only the right arm on the full mesh.

In comparison, the quadric mesh simplification method as proposed in [26] and [7] doesn't provide such local semantic control in its latent space, as it simplifies the mesh according to the point to plane error of a template, so the receptive field is prone to respect Euclidean space rather than geodesic distance. In Figure 2b, one can see the receptive fields from the quadric mesh simplification to both the right arm and the right hip, which is less favorable for localized interpolation.

## 4 Experiments

In this section, We first examine the generality of our AE models on different types of 3D mesh data. Then, we present localized latent code interpolation for 3D hand models. After that, we compare our model with SOTA 3D mesh AEs. Finally we compare the performance of different convolution and (un)pooling layers under the same network architecture. All experiments were trained with L1 reconstruction loss only, Adam [17] optimizer and reported with point to point mean euclidean distance error if not specified. Additional experiment details can be found in the appendix.

### 4.1 Generality

We first experimented on the 2D-manifold D-FAUST human body dataset [4]. It contains 140 sequences of registered human body meshes. We used 103 sequences (32933 meshes in total) for training, 13 sequences (4023 meshes) for validation and 13 sequences (4264 meshes) for testing. Our AE has a mean test error at 5.01 mm after 300 epochs of training.

Then we tested for the 3D-manifold cases using 3D tetrahedrons (tet meshes) ( Figure 3b). Tet meshes are commonly used for physical simulation [3]. We used a dataset containing 10k deformed tet meshes of an Asian dragon and use 7k for training an AE, 1.3k for validation and 562 for testing. After 400 epochs of training, the error converged to 0.2 mm.

To demonstrate our model on non-manifold data, we trained the network on a 3D tree model with non-manifold components [3] (Figure 3c). We made a dataset of 10 sequences of this tree's animation simulated by Vega using random forces, 1000 frames for each clip, and used 2 clips for testing, 2 clips for validation and the rest for training. After 36 epochs, the reconstruction error dropped to 4.1 cm.

3D data in real applications can have very high resolution. Therefore, we experimented our network on a high-resolution human dataset that contains 24,628 fully aligned meshes, each with 154k vertices and 308k triangles. We randomly chose $2,463$ for testing and used the rest for training. The AE's bottleneck contains 18 vertices and 64 dimensions per vertex, resulting in a compression rate of 0.25%. After training 100 epochs, the mean test error dropped to 3.01 mm. From Figure 3a we can see that the output mesh is quite detailed. Compared with the groundtruth, which is more noisy, the network output is relatively smoothed. Interestingly, from the middle two images, we can see that the network learned to reconstruct the vein in the inner side of the arm which is missing from the originally noisy surface.

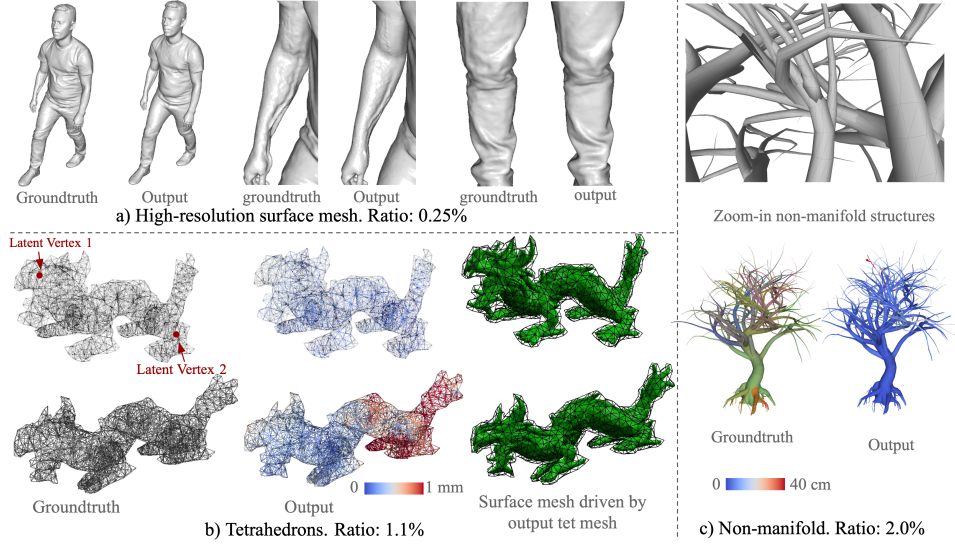

a) High-resolution surface mesh. Ratio: 0.25%

Zoom-in non-manifold structures

b) Tetrahedrons. Ratio: 1.1%

Surface mesh driven by output tet mesh

c) Non-manifold. Ratio: 2.0%

Figure 3: Reconstruction results for different meshes and compression ratios.

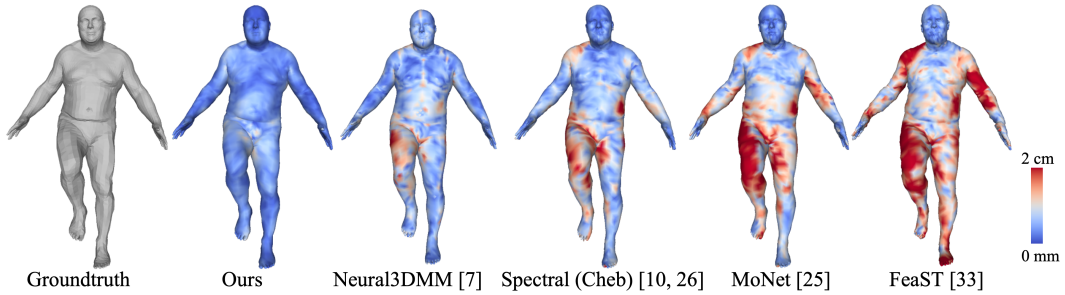

Groundtruth      Ours      Neural3DMM [7]      Spectral (Cheb) [10, 26]      MoNet [25]      FeaST [33]

Figure 4: Reconstruction results of using different mesh convolution networks.

## 4.2 Localized Latent Feature Manipulation

We demonstrate localized latent feature interpolation with an AE trained on hand meshes. As in Figure 6, we set the latent vertices to be at the tips of the five fingers and the wrist. For interpolation, we first inferred the latent codes from a source mesh and a target mesh, then we linearly interpolated the latent code on each individual latent vertex separately. With only two input hand models, we can obtain many more gestures by interpolating a subset of latent vertices instead of the entire code.

We further demonstrate the reconstruction result by mixing local latent features of two D-FAUST models. As shown in Figure 5, we replace "Man A's" latent code which corresponds to the foot area with "Man B's" foot latent code. The result is Man A with a new foot pose, while keeping the rest of Man A the same. In particular, note that Man A's new leg pose is still "his" leg, and not "Man B's" leg. This shows that we can in fact perform reasonable pose transfer even using different identities.

## 4.3 Comparison of 3D Mesh Autoencoder Models

For comparison, we choose to test on the D-FAUST dataset as it captures both high-frequency variance in poses and low-frequency variance in local details and is widely used for estimation in previous works.

We compare our autoencoder with Neural3DMM [7] which is the current SOTA work for registered 3D meshes. Both networks are set with compression ratio (network bottle neck size over original size) at 0.3%. We trained ours with 200 epochs and Neural3DMM with 300 epochs. As reported in Table 1, our network achieves over 30% and 40% lower errors for the testing and training set

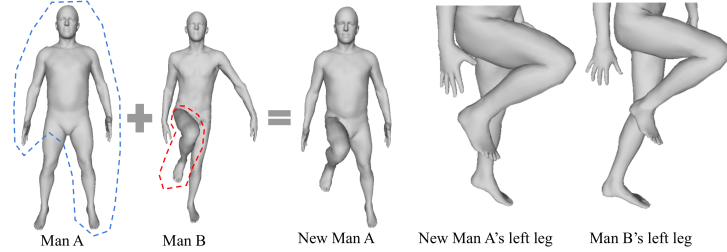

Figure 5: Mixing latent codes from two shapes.

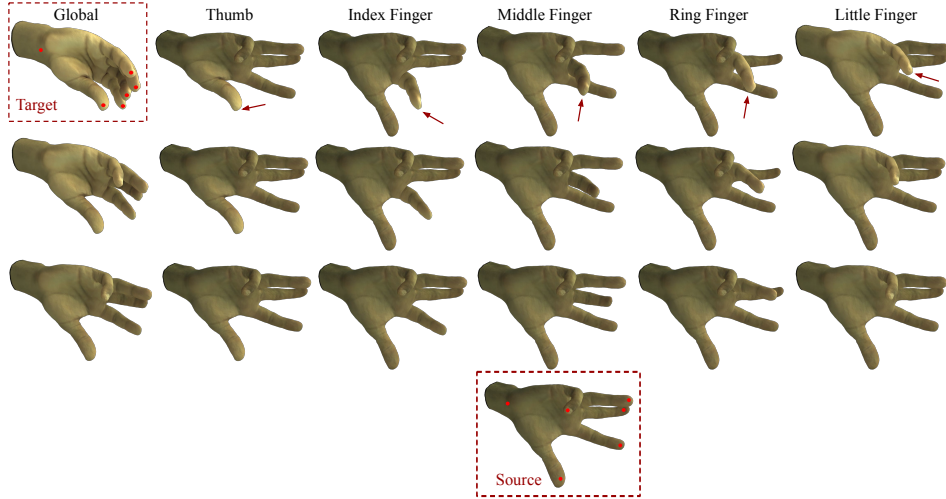

Figure 6: Interpolation from source to target using global or local latent codes.

respectively with fewer training epochs and learning parameters. A visual comparison between the two methods can be found in Figure 4.

We also compare with MeshCNN [14], but found that MeshCNN is infeasible for the full resolution D-FAUST dataset due to memory and speed constraints. Thus, we have to down-sampled all meshes to 750 vertices, which is the same size used in their experiments. Because [14] performs convolution on the edges, we use each edge's two endpoints as the input feature we attempt to reconstruct. We set the input size to 3000 edges and have a bottleneck layer of 150 edges. The number of layers and channels is the same as in [14]. With 200 epochs of training, this network still doesn't perform well, as reported in Table 1.

## 4.4 Comparison of Network Layers

In this section, we evaluate our proposed network with other design choices. Based on the architecture defined in Section 3.3, we keep the block number, the input/output channel and vertex number the same (Except for GATConv in Experiment 2.5 which has eight times more channels), but compare with different convolution and (un)pooling layers. All experiments were trained on the D-FAUST dataset with the same setting. Table 1 lists all the block designs, the errors, the parameter count and the GPU memory consumption for training with batch size=16.

In Group 1 Ablation Study, we compare different attributes and combinations of our proposed layers. From 1.1 to 1.3, we test the effect of adjusting $M$. We denote the encoding or decoding residual blocks as vcDown/TransConv(s, r, $M$) + vcDown/UpRes(s). By increasing $M$, the network's capacity increases and achieves lower errors. In 1.4 the residual layers are removed and errors go higher. In 1.5 we replace the convolution and transpose convolution by the combination of $s = 1$ convolution and $s = 2$ pool or unpool, denoted as vcConv->vdPool and vdUnpool->vcTransConv and the errors increase a bit. In 1.6 we apply normalization on the weight bases and the error further increases.

| | Error (mm) | | |
|---|---|---|---|
| | Train | Test | Param |
| Ours | 3.73 | 5.01 | 1.9m |
| Neural-3DMM | 6.42 | 7.39 | 2.0m |
| Mesh-CNN* | 83.3 | 101.7 | 2.2m |

0. Comparison of Whole Models

| | Encoder Block | Decoder Block | Error (mm) | | Param | Train Mem |
|---|---|---|---|---|---|---|
| | | | Train | Test | | |
| 1.1 | vcDownConv(2,2,37) + vcDownRes(2) | vcTransConv(2,2,37) + vcUpRes(2) | **3.02** | **4.56** | 3.9m | 2509Mib |
| 1.2 | vcDownConv(2,2,27) + vcDownRes(2) | vcTransConv(2,2,27) + vcUpRes(2) | 3.29 | 4.73 | 2.9m | 2493Mib |
| 1.3 | vcDownConv(2,2,17) + vcDownRes(2) | vcTransConv(2,2,17) + vcUpRes(2) | 3.73 | 5.01 | 1.9m | 2471Mib |
| 1.4 | vcDownConv(2,2,17) | vcTransConv(2,2,17) | 4.02 | 5.23 | 1.8m | 2123Mib |
| 1.5 | vcConv(1,1,9) - vdPool(2) | vdUnpool(2)-vcConv(1,1,9) | 4.57 | 5.63 | 1.4m | 4183Mib |
| 1.6 | vcConv(1,1,9)* - vdPool(2) | vdUnpool(2)-vcConv(1,1,9)* | 13.25 | 14.29 | 1.4m | 4183Mib |

1. Ablation Study

| | Encoder Block | Decoder Block | Train | Test | Param | Train Mem |
|---|---|---|---|---|---|---|
| 2.1 | vcConv(1,1,9) - vdPool(2) | vdUnpool(2)-vcConv(1,1,9) | 4.57 | 5.63 | 1.38m | 4185Mib |
| 2.2 | LCConv(1,1) - vdPool(2) | vdUnpool(2)-LCConv(1,1) | **2.67** | **4.23** | 185.7m | 8767Mib |
| 2.3 | ChebConv(1,1) - vdPool(2) | vdUnpool(2)-ChebConv(1,1) | 7.19 | 8.59 | 0.15m | 1853Mib |
| 2.4 | MoNetConv(1,1) - vdPool(2) | vdUnpool(2)-MoNetConv(1,1) | 9.21 | 10.4 | 0.61m | 5223Mib |
| 2.5 | GATConv(1,1) - vdPool(2) | vdUnpool(2)-GATConv(1,1) | 11.95 | 14.28 | 0.21m | 7377Mib |
| 2.6 | FeastConv(1,1) - vdPool(2) | vdUnpool(2)-FeastConv(1,1) | 14.77 | 17.03 | 0.76m | 7359Mib |

2. Comparison of Convolution Layers

| | Encoder Block | Decoder Block | Train | Test | Param | Train Mem |
|---|---|---|---|---|---|---|
| 3.1 | vcConv(1,1) - vdPool(2) | vdUnpool(2)-vcConv(1,1) | **4.57** | **5.63** | 1.38m | 4185Mib |
| 3.2 | vcConv(1,1) - avgPool(2) | avgUnpool(2)-vcConv(1,1) | 5.93 | 6.88 | 1.37m | 3651Mib |
| 3.3 | vcConv(1,1) - maxPool(2) | maxUnpool(2)-vcConv(1,1) | 8.8 | 13.55 | 1.37m | 3651Mib |
| 3.4 | vcConv(1,1) - qPool(2) | qUnpool(2)-vcConv(1,1) | 4.94 | 6.08 | 1.37m | 4731Mib |

3. Comparison of Pooling Layers

Table 1: Comparison of whole models and layers on D-FAUST dataset.

In Group 2, we compare our vcConv with other convolution operators as proposed in previous works. Since all the other convolution layers don't support up or down-sampling, we use our vdPool and vdUnpool layers for sampling, and set $s = 1, r = 1$ for all convolution layers. From the table, 2.2 LCConv layer, which never learns any shared weights, has the lowest error but it has around 135 times more learnable parameters than our vcConv layer and consumes twice the GPU memory for training, preventing it from being applied to bigger network architectures, like with high-resolution meshes. In 2.3, spectral convolution with 6 Chebyshev bases [10] has the least parameters but the test error increases 50%. For 2.4 MoNet [25], we use the pseudo coordinates from its paper, and set the kernel size to 25; For 2.5 GATConv [32], we set the head number being 8, all heads concatenated except for the middle and last layer which were averaged instead; for 2.6 FeastConv [33], we set the head size as 32. 2.3 to 2.6 were implemented with PyTorch Geometry [13]. They have much higher error than our vcConv and require much more memory. This demonstrates our method achieves the best accuracy and efficiency in terms of memory consumption. A visual comparison among using vcConv, Spectral, MoNet and FeaST convolution layers can be found in Figure 4.

In Group 3, we keep the vcConv layers the same but use different (un)pooling layers. From 3.2 and 3.3, we can see that using simple average or max (un)pooling operations increases the error. In 3.4, using the quadric layers (qPool), the error also increases.

# 5 Conclusion

We introduce a novel mesh AE that produces SOTA results for regressing arbitrary types of registered meshes. Our method contains natural analogs of the operations in classic 2D CNNs while avoiding the high parameter cost of naive locally connected networks by using a shared kernel basis. It is also the first mesh AE that demonstrates localized interpolation.

Several future directions are possible with our formulation. For one thing, though our method can learn on arbitrary graphs, all the graphs in the dataset must have the same topology. Thus, it requires that the mesh connectivity is already solved in the training set. In the future, we plan to extend our work so that it can work on datasets with varying topology.

# 6 Potential Broader Impact

This work can potentially impact future entertainment and communication industry. It could also allow for more efficient storage and transport of 3D data.

# 7 Funding Disclosure

Additional revenues related to this work: Chenglei Wu, Chen Cao, Yuting Ye, Jason Saragih and Yaser Sheikh at Facebook Reality Labs; Yi Zhou at Adobe, previously at the University of Southern California and did internship at Facebook Reality Labs; Zimo Li at the University of Southern California; Hao Li at Pinscreen and previously at the University of Southern California and USC Institute for Creative Technologies.

## Footnotes

[3]Model is from `https://www.turbosquid.com/3d-models/cherry-blossom-tree-3d-1189864`

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
