[Supplementary Material]



Figure 5: Steps for graph up and down-sampling.

# A   Method Details

## A.1   Graph Sampling Algorithm

To select a subset of sampled vertices from the original graph with stride $s$, for each connected component, we start from a random vertex, mark it as selected and mark its $(s-1)$-ring neighbors as removed, and then traverse the vertices on the $s$-ring until finding a vertex that is not marked and has no (s-1)-ring neighbors marked as selected. Next, we mark this vertex as selected and start a new round of searching. We repeat those steps recurrently using a queue structure until all vertices are marked. One can also manually assign certain vertices to be included in the sampling set by marking them before starting the searching.

For down-sampling, given stride $s > 1$, as shown in Figure 5, $\{x_i\}$ are the blue vertices with orange circles sampled with $s = 2$. In the output graph $\mathcal{Y}$, we create a vertex $y_i$ (yellow vertices) for each $x_i$. The topology of $\mathcal{Y}$ purely depends on the vertex connectivity in $\mathcal{X}$: two vertices are connected in $\mathcal{Y}$ if their distance is less than or equal to $2s - 1$ in $\mathcal{X}$. With a kernel radius of $r$, $\mathcal{N}(i)$ contains the $r$-ring neighborhood of $x_i$.

For up-sampling, the input and output graphs are the output and input graphs in down-sampling with the same stride size. It's a dual process of down-sampling. To determine $\mathcal{N}(i)$ of $y_i$ for a kernel radius $r$, we first collect $y_i$'s $r$-ring neighbors in $\mathcal{Y}$, then locate the sampled vertices (yellow-circled) and include their corresponding vertices in $\mathcal{X}$ to construct $\mathcal{N}(i)$.

## A.2   Network Implementation

In our implementation, we precompute the graph sampling process and record $\{\mathcal{N}(i)\}$ in a table of vertex connections. Each line $i$ in the table contains the indices of the vertices in $\{\mathcal{N}(i)\}$ in the input graph. In practice, we store the table using a $N \times Max(E_i)$ integer tensor. The training parameters for convolution coefficients are stored in a tensor of size $N \times Max(E_i) \times M$ and the basis is a tensor of $M \times O \times I$. We use a $N \times Max(E_i)$ mask to mask out the vacant entries.

All of our networks in Section 4.1 and 4.2 are composed of residual blocks illustrated in Figure 6

For training and testing, the whole network forward and backward processes are fully parallelized in GPU written in Pytorch. During testing, the kernels are pre-computed using equation 2 to accelerate the inference time. All networks are trained with batch size=16, learning rate=0.0001, learning rate decay=0.9 every epoch, using Geforce 1080Ti, cuda 10.0 and pytorch 1.0.

Figure 6: Residual block for down/upsampling. (vd should be vc)

Figure 7: Artifact along seam lines in UV mapping method.

## B  Experiment Details

The dragon tet mesh[2] in Section 4.1 has 959 vertices and lies in a bounding box of roughly $20{\times}20{\times}20$ cm. The auto-encoder has five down-sampling residual blocks and five up-sampling residual blocks. $s = 2, r = 2, M = 37$ for all blocks. Its bottle neck has two vertices, 16 latent codes per vertex.

The 3D tree model[3] has 328 disconnected components. To connect all the components, we added an edge between each pair of close components. The auto-encoder had four down-sampling residual blocks to compress the original 20k vertices into 149 vertices, 8 latent codes per vertex and four up-sampling residual blocks. $s = 2, r = 2, M = 27$ for each block.

For the high resolution human dataset, Our auto-encoder has 6 down-sampling residual blocks and 6 up-sampling residual blocks, with $s = 2, r = 2$ and $M = 17$ for all blocks. The latent space has 18 vertices and 64 dimensions per vertex, resulting in a compression rate of 0.75%. We additionally used L1 laplacian loss for training.

The hand dataset contains fully aligned hand meshes reconstructed from performance captures of two people with roughly 200 seconds of 90 poses per person. The mesh has 57k vertices and 115k facets. We randomly picked 39 one-second clips for testing, 39 one-second clips for validation, and used the rest for training, resulting in 9376 meshes for training and 1170 for testing. Each vertex is given both the 3D coordinate and RGB color as input. Our auto-encoder network has nine down-sampling residual blocks and nine up-sampling residual blocks with $s = 2$, $r = 2$ and $M = 17$ for all blocks. The latent space has 6 vertices with 64 dimensions per vertex, resulting in a compression rate of 0.11%. The middle 6 vertices were manually selected to be at the fingertips and wrist. We trained the network with point to point L1 loss, L1 laplacian loss and L1 RGB loss. After 205 epochs of training, the mean point to point euclidean distance error dropped to 1.06 mm and the mean L1 RGB error to 0.036 (range 0 to 1).

## C  Additional Figures

Figure 7 shows the artifact along seam lines in UV mapping based mesh CNN methods.

## Footnotes

[2]The original mesh was created by Stanford Computer Graphics Laboratory, available at `http://graphics.stanford.edu/data`, and then simulated using Vega [3].

[3]The original model is downloaded from `https://www.turbosquid.com/3d-models/cherry-blossom-tree-3d-1189864`