[Reviews · NeurIPS 2020]

Review 1

Summary and Contributions: This work proposes a novel template-free fully convolutional auto encoder for arbitrary registered meshes. A spatially-varying convolution kernel is used to deal with for irregular sampling and connectivity. It performs experiments on D-FAUST to show the advantage over existing methods.

Strengths: The major novelty is a mesh AE for arbitrary topology which captures effectively captures local geometry property. Decomposing weights to vertex-specific weighted basis functions is the key to success. Experiments are thorough and convincing.

Weaknesses: Is the local neighborhood N defined based on Euclidean distance or geodesic distance? It is not clear.

Correctness: Yes.

Clarity: Yes.

Relation to Prior Work: Yes.

Reproducibility: Yes

Additional Feedback:


Review 2

Summary and Contributions: The paper proposes a fully convolutional mesh autoencoder. The key ideas include a spatially varying kernel formed by a linear combination of basis kernels, as well as a more adaptive pooling operation. The method shows improved performance for autoencoder based mesh reconstruction.

Strengths: The ideas are generally plausible. The method outperforms state-of-the-art Neural 3DMM.

Weaknesses: While using spatially varying kernels may be new for meshes, the idea has been well studied in image-based CNN, and the extension seems fairly straightforward. The paper should refer to the image-based work, e.g. Adaptive Convolutional Kernels, ICCV Workshop 2019. and clarify the contribution of the paper . Comparison with MeshCNN in this paper is a bit unfair as the method is not designed for reconstruction. Other than this, the paper only compares with one method, and could benefit from comparison with more state-of-the-art methods to be convincing.

Correctness: The method seems generally correct. The paper claims that the capability of handling non-manifold etc. is a strength of the method. However, existing methods based on graph CNN can also be applied to non-manifold cases.

Clarity: The writing is fine.

Relation to Prior Work: As discussed above, the key idea of using spatially varying kernels has been considered in 2D CNN, so this should be discussed: Adaptive Convolutional Kernels, ICCV Workshop 2019.

Reproducibility: Yes

Additional Feedback: Post-rebuttal comments: The rebuttal has promised to make the requested changes, so I will keep the positive score with the paper.


Review 3

Summary and Contributions: The paper proposes a new convolution technique for graphs that share the same connectivity. It is based on learning a global "imaginary" linear basis for the convolutional filters. Every neighborhood can define its own kernel by learning coefficients for this basis. This permits to have an entirely convolutional network, giving also interpretability of the latent representation.

Strengths: Contribution ========== I think the proposed technique is novel and it proposes a different point of view. I appreciated the effort of defining all the needed layers (Pooling, UnPooling, UpRes, DownRes). I find it really interesting the idea to have an "imaginary basis"; it is somehow related to learning the Gaussian parameters of [25], but with more degrees of freedom. This idea can be easily extended to several representations as shown in the paper (tetrahedrons, non-manifold mesh). I was wondering: would it be possible to extend it on point clouds, e.g. by using a Gaussian distance to define the neighborhood? Localized Latent feature Interpolation ======== I think this is a nice point: the capability to have an interpretable latent representation is a hot topic in this research field. While in the video of the hands there are some unnatural artifacts, in Figure 2 the latent representation looks like a skeleton and it seems there are potentially nice connections with standard skinning systems. I would love to see some interpolation examples between humans, in both pose (i.e. same subject in different poses) and identity (i.e. same pose with different subjects) interpolations.

Weaknesses: Geometry ======== From a geometrical point of view, this method takes into account only the connectivity; the shapes are seen as a graph with features (i.e. the 3D coordinates). Also, the vertices need to be ordered, i.e. it assumes to have fixed connectivity in input. I think these are the two major drawbacks. Working with meshes, several applications (e.g. point-to-point matching) cannot be addressed. For example, [25] does not suffer from this problem. Can you think of any strategy to overcome these limits? I also think that setting the M parameters relies on a further assumption that the connectivity is almost evenly distributed on all the graphs. How does the method perform in case of high-variance in the connectivity? Experiments ======== More details in the "Correctness" box

Correctness: The paper seems correct to me, from a theoretical perspective. I have only one point about learning the Weight Basis and locally variant coefficients at the same time; it looks like a min-min problem (e.g. finding the best coefficient for the basis, and find the best basis for the coefficients) and in general it is not directly optimizable. I am wondering if sometimes it rises some undefined solutions during training, e.g. basis collapsing in zero-vectors. Have you experienced any issues with this? A visualization of the learned imaginary basis would be definitely useful. However, my major concern is on the experimental setup. The ablation study is extensive, but the method has been compared with only two previous works, in only one task. No baseline has been provided, and there is a lack of applications (e.g. Analogies, Extrapolation, Latent space samplings present in [7]). I find a bit disappointing that the capability of localized interpolation has so few space in the paper. I would see more on the semantic capability of the learned representation.

Clarity: I think the paper have some issues to fix. 1) As stated in the "Prior work" box, I find the introduction and related work a bit repetitive. I would suggest merging the two sections 2) Table 1 is a bit too small in the printed version, and I think could be better organized. 3) I would rename section 4.4 as "Ablation study", for clarity 4) It is a bit hard to appreciate the trees in Figure 3. Also, it is not super evident how they are non-manifold. I would suggest having a larger figure just for them, maybe with some zoomed detail 5) I would love to have a visual insight into the learned Weight Basis. 6) I would suggest changing the word "topology" with "connectivity"; the first means several different things and could be misleading. Minor typos: - row 41: space missing after "data." - row 53: "is the use a spatially-varying" -> "is the use of a spatially-varying" - row 127: "Tasks that requires" -> "Tasks that require"

Relation to Prior Work: I think the prior works are well discussed and I have no comment on this. I find only a bit repetitive the introduction and section 2.2; I would suggest merging the two sections to ease the reading.

Reproducibility: Yes

Additional Feedback: 1) The weight basis B are shared for all the mesh (i.e. all the neighborhoods use the same basis set). Then, the M parameter is a fixed scalar for all the neighborhoods, and I think it is around 8-10 for a genus-0 trimesh. Is it correct? I think this is a useful detail. Also: does highly non-uniform connectivity requires special effort? 2) I have not found any mention of your implementation. Are you planning to release it in the future? 3) Since you learn some imaginary basis, they also could be non-linear (e.g. using an hypernetwork setup). There is some advantage to use a linear combination of the learned basis? Also, have you ever had problems with the linear independence of the learned basis? 4) Shouldn't Equation 1 and 3 multiplications between W_j and x_{i,j} be switched to match the dimensionalities (Since x_{i,j} is \in R^I and W_j \in R^{I \times O})? 5) In the "Conclusion" paragraph is stated that "We plan to extend our work in the future so that it can work on datasets with varying topology". Could you elaborate on this point, especially if there are some specific directions to investigate? This extension does not seem straightforward to me, and actually, it is the major limitation of the architecture. Rebuttal Comment ================= I thank the authors for their rebuttal and their effort to reply to my questions. I still have some concerns: - Provided interpolation example is a “Latent Algebra” example. While it still interesting and I appreciated it, it does not provide an insight into how the intermediate frames look (i.e., semantic coherence between poses). Also, the rendering makes it difficult to understand the impact on the rest of the body - In the rebuttal is stated that adding regularizations hurts performance. If authors already performed these experiments, I think would be worthful to include them in the paper. Also, no further study has been performed on the learned basis; it is a pity, and would be interesting to provide some insight on this. - Rebuttal confirms that the work is placed in contexts where the connectivity (and then the correspondence) among the training set is already solved. This limits the applicability of the method (e.g. shape matching, segmentation). Concluding, while I think the paper still have some experimental\analysis issues, I appreciate the idea and I think it is interesting for the NIPS audience. For these reasons, I will keep my initial vote.

[Author Response · NeurIPS 2020]


Groundtruth    Ours    Neural3DMM [7]    Spectral (Cheb) [10, 26]    MoNet [25]    FeaST [33]

Man A    Man B    New Man A    New Man A's left leg    Man B's left leg

We appreciate the reviewers' feedback. We will fix the typos and errors, add the ICCV Workshop 2019 paper (**R2**) within the related work and open-source our code. Below, we list reviewer concerns in bold and address them underneath.

**Concern over experimentation (R2 and R3)** For the concern of not enough comparisons with competitor methods, while it is true that we compare only with 2 other "entire" methods (Table 1.0), in fact we compare with 5 additional graph convolution networks in Table 1.2, resulting in a total of 7 comparisons. The distinction is because the methods in Table 1.2 come from publications which do not provide up and down-sampling schemes and therefore cannot perform the autoencoder task, so it is not possible to compare with them in an "entire" fashion. The only way to compare with them is to use their proposed convolution layers inside our pipeline, which is designed for the task. To alleviate the confusion for this, we will add citation links next to methods listed in Table 1.2, so that it is clear they are competitor methods. In addition, we will add error visualizations of these methods (Top half of added figure). Finally, we are also happy to add any additional comparisons with baselines or competitor methods the reviewers wish to see, as well as further interpolation, extrapolation, and analogy examples to the final appendix.

**"I would love to see some interpolation examples between humans..."(R3)** Bottom half of added figure shows identity of "Man A" while interpolating only the latent-node which corresponds to the foot area of "Man B". The result is Man A with a new foot pose, while keeping the rest of Man A the same. In particular, note that Man A's new leg pose is still "his" leg, and not "Man B's" leg. This shows that we can in fact perform reasonable pose interpolation even using different identities. We can add additional human interpolation to the final appendix if desired.

**"The vertices need to be ordered...it assumes to have fixed connectivity in input... [25] does not suffer from this problem...Can you think of any strategy to overcome these limits?" (R3)** It is true that the requirement of fixed-connectivity datasets is the main limitation of our method. However, using registration to get the same connectivity across dynamic meshes is well studied, and high accuracy correspondence can be achieved. Therefore, it is not necessary to solve the connectivity and shape problem simultaneously. While [25] gets around this, its accuracy and reconstruction fidelity is much worse than fixed-connectivity methods like ours or Neural3DMM [7] (see the added figure, top). It is our impression that there is a clear trade-off here between reconstruction quality and generalize-ability, and we are examining the other side of the tradeoff compared to [25]. As to strategies which allow our convolution method to extend to variant topology input while keeping the reconstruction fidelity high, we believe this is a quite difficult problem (middle of the tradeoff-curve) and will require several future research projects.

**"It looks like a min-min problem...and in general it is not directly optimizable." (R3)** Actually, it is not necessary to do any sort of alternating optimization. In fact, our formulation of convolution through the basis and coefficients is a fully differentiable operation, and so all parameters can be jointly optimized with standard SGD. We have also not encountered any issues with basis vectors going to 0 during training.

**"...[is] There is some advantage to use a linear combination of the learned basis? Also, have you ever had problems with the linear independence of the learned basis?" (R3)** The linear option is the straightforward choice. We have encountered no problems with linear independence of the learned basis, though it is possible to put in some type of regularization to prevent correlation between basis vectors. In early experiments, however, we have found adding regularization to the basis hurts performance.

**"...assumption that the connectivity is almost evenly distributed on all the graphs. How does the method perform in case of high-variance in the connectivity?" (R3)** The method has no requirement on the distribution of connectivity. The tree example (Figure 3) does not have an even distribution of connectivity, and the network is able to learn it well. We can add zoomed-in figures of the tree example to show it's uneven connectivity.

**Is the local neighborhood N defined based on Euclidean distance or geodesic distance? (R1)** It is defined on vertex connectivity neighborhood.

**"...should multiplications between W j and x ij be switched?" (R3)** Yes, thank you, it should be $W_j^T$

[Meta-Review · NeurIPS 2020]

All the reviewers voted for the paper to be accepted and praised the novelty of the proposed mesh autoencoder. They are however expecting the changes promised in the rebuttal to be included in the camera ready version of the paper. They also suggest stating clearly in the paper the limitations -- namely that connectivity is already solved in the training set, which limits applicability of the method.